# Airborne Dental Material Particulates and Occupational Exposure: Computational and Field Insights into Airflow Dynamics and Control Strategies

**DOI:** 10.3390/toxics13110957

**Published:** 2025-11-05

**Authors:** Chanapat Chanbandit, Kanchana Kanchanatawewat, Ghaim Man Oo, Jatuporn Thongsri, Kuson Tuntiwong

**Affiliations:** 1Computer Simulation in Engineering Research Group, Department of Manufacturing System Technology, School of Integrated Innovative Technology, King Mongkut’s Institute of Technology Ladkrabang, Bangkok 10520, Thailand; chanbandit.cha@gmail.com (C.C.); ghaimman.oo@kmitl.ac.th (G.M.O.); 2School of Dentistry, King Mongkut’s Institute of Technology Ladkrabang, Bangkok 10520, Thailand; kk_tawewat@yahoo.com

**Keywords:** aerosol mitigation, Box Dust Collector (BC), Computational Fluid Dynamics (CFD), dental material particles, occupational safety, portable air purifiers (PACs)

## Abstract

Occupational exposure to airborne polymethacrylate (PMMA) particles during dental laboratory procedures poses an underexplored health risk. This study presents the first integrated Computational Fluid Dynamics (CFD) and real-time particle monitoring investigation of 0.5 µm PMMA particle dispersion during mechanical polishing in an actual clinic. We quantitatively assessed particle behavior in 30 s exposure scenarios by examining the effects of dental professional work orientations and comparing two mitigation strategies, rear-inlet portable air cleaners (PACs) and a Box Dust Collector (BC), with an emphasis on the safety of both personnel and patients. The findings establish that operatory airflow is a primary safety determinant: aligning the workflow with the main airflow (0°). Furthermore, the combined use of PACs and BC demonstrated synergistic superiority, achieving the optimal reduction in peak concentrations and airborne residence time. PACs alone reduced working zone concentrations by up to 80%, while BC provided a crucial 40–60 s delay in initial plume dispersion. We conclude that effective exposure control requires a proactive, two-stage engineering defense: source confinement augmented by continuous ambient filtration. This research provides a robust, evidence-based foundation for defining airflow-aware ergonomic and combined engineering standards in the evolving digital era of dentistry.

## 1. Introduction

In recent years, the issue of airborne particulate matter (PM), particularly particles smaller than 2.5 μm (PM2.5), has garnered increasing global attention [1]. According to reports from the World Health Organization (WHO) and numerous studies in environmental and health sciences, long-term exposure to PM2.5 is associated with increased mortality from respiratory diseases, heart disease, and cardiovascular conditions [2,3,4,5]. These fine particles can penetrate deep into the bronchi and alveoli, where they may accumulate in lung tissue and potentially trigger chronic inflammation [6,7]. Although the human body can naturally eliminate some of the accumulated particles, the process is slow [8,9]. Given the current high density of ambient PM2.5, the health risks cannot be overlooked. Medical data further indicate that approximately 10–20% of patients with chronic respiratory illnesses have a history of prolonged exposure to high particle concentrations [10].

Despite widespread awareness of these health risks, some occupational groups continue to face unavoidable exposure, especially those working in enclosed environments with limited ventilation, such as industrial facilities, office buildings, and healthcare settings. One particularly high-risk but often overlooked area is the dental clinic, where numerous procedures generate large quantities of fine particles [11]. Dental practice represents a unique occupational environment characterized by the complex interplay of biological aerosols, chemical vapors, and solid particulates. While extensive research has focused on the infectious risks associated with bioaerosols generated during aerosol-generating procedures (AGPS), such as tooth preparation or ultrasonic scaling [12,13,14,15], comparatively limited attention has been directed toward non-biological airborne particulates arising from the adjustment and polishing of dental materials within the clinical setting. These non-biological particulates originate from commonly used materials—including occlusal splints, surgical stents, implant guides, provisional restorations, and 3D-printed components—that are typically fabricated in dental laboratories but frequently refined chairside during patient treatment. The processes involved include high-speed grinding, pressurized air or water sprays, and the grinding or polishing of dental materials [16,17,18]. Particles produced during these processes often contain a complex mixture of biological dust, chemicals, and material-derived components, which can remain suspended in the air from 10 min to over an hour, depending on ventilation conditions [19]. Consequently, dental professionals are at heightened risk of inhaling airborne contaminants during their procedures [20,21,22,23,24].

The evolution of modern dentistry toward digital and decentralized workflows has facilitated the widespread integration of in-house dental laboratories and on-site fabrication techniques, including computer-aided design/computer-aided manufacturing (CAD/CAM) milling systems and additive manufacturing (3D printing) [25,26,27]. These technological innovations enhance treatment efficiency and patient convenience while reducing turnaround times; however, they inadvertently introduced new occupational hazards that remain inadequately characterized [28]. Recent investigations have documented compositional profiles of particulate matter generated during dental procedures. Tang et al. [29] reported elevated concentrations of potentially toxic elements including cadmium, arsenic, and nickel, in PM2.5 samples collected from multi-chair dental clinics, with carcinogenic risk estimates exceeding acceptable safety thresholds. Ding et al. [30] highlighted this critical yet often overlooked aspect: the hazards associated with the release of fine particulate matter during the grinding or polishing of dental devices, which generate numerous fine particles that may trigger respiratory inflammation or disease. Epidemiological evidence suggests that chronic exposure to dental material particulates may contribute to respiratory complications among dental professionals. Studies have documented associations between occupational dust exposure in dental settings and increased prevalence of pulmonary symptoms. However, the specific contributions of material-derived particulates versus biological aerosols remain incompletely elucidated [31]. The spatial distribution of these airborne particles—governed by complex interactions between local airflow patterns, ventilation design, equipment positioning, and the dynamic movements of dental personnel—remains poorly understood [32,33]. This knowledge gap significantly limits the development of evidence-based engineering controls and exposure mitigation strategies specifically tailored to contemporary dental practice environments.

This spatial distribution of these airborne particles—governed by complex interactions between local airflow patterns, ventilation design, equipment positioning, and the dynamic movements of dental personnel—remains poorly understood. This critical knowledge gap significantly limits the development of evidence-based engineering controls and exposure mitigation strategies tailored explicitly to contemporary dental practice environments. This lack of data becomes acutely critical given the rapid expansion of modern dentistry toward in-house dental lab and on-site fabrication (digital manufacturing within the clinic), which inadvertently introduces new emission sources. Our study fundamentally differentiates itself by focusing on airborne particulates originating from dental materials, PMMA, resins, and zirconia during mechanical fabrication. Unlike bioaerosols, these synthetic particles present a distinct long-term toxicological risk. Our research, therefore, intentionally shifts the analytic paradigm from acute infectious control toward the engineering solutions required for mitigating this material-specific occupational burden, stressing the profound importance of particulate control in these high-risk in-house fabrication zones.

### 1.1. Computational Fluid Dynamics (CFD) Applications in Dental Settings

Previous studies have employed CFD models to investigate the dispersion of bioaerosols in controlled environments, including fever clinics and dental clinics. Zhou and Ji [34] and Liu et al. [35] focus primarily on assessing infection risk and identifying areas of particle accumulation. In 2022, Liu et al. [36] extended this work by evaluating aerosol behavior and calculating the probability of infection in the working area of dental practitioners across multiple clinic rooms, thereby identifying high-risk areas for SARS-CoV-2 transmission. Although these studies provide a clear overview of health risks posed by bioaerosols, they predominantly focus on pathogens and aerosols generated during oral treatment procedures, with limited evaluation of the influence of airflow patterns on dentists’ working positions and clinic equipment layouts—factors that critically affect particle dispersion and accumulation in enclosed spaces. Recent advances in CFD analysis have enabled sophisticated simulation of complex airflow patterns and particle dispersion phenomena within the dental clinical environment [37]. When validated through real-time particle measurements and experimental verification, these computational approaches offer unprecedented insights into how airborne particulates behave, accumulate, and respond to different mitigation measures under realistic working conditions [38]. CFD modeling has proven particularly valuable for evaluating the effectiveness of engineering controls, including local exhaust ventilation, extraoral suction systems, portable air purification devices, and optimized room ventilation strategies [39].

### 1.2. Engineering Controls and Mitigation Strategies

Regarding engineering controls, studies by Chen et al. [40], Kashkooli et al. [41], Novoselac & Siegel [42], and Dai and Zhao [43] have demonstrated that the position of portable air purifiers (PACs) significantly impacts their air-cleaning effectiveness. However, these studies were conducted in general-use settings, such as living rooms or offices, and focused on pollutants like cigarette smoke, water vapor, or PM2.5. The aerodynamic behavior and emission characteristics of dental particulates differ considerably from those of common pollutants. Importantly, no previous studies have examined or compared the effect of PAC inlet–outlet airflow direction on local airflow patterns in the critical working area, particularly the dentist’s breathing zone during treatment. As a result, it remains unclear which PAC design is most effective in reducing aerosol deposition in these areas. CFD-based optimization studies have further demonstrated that strategic modifications to ventilation layouts and increased air change rates can markedly improve contaminant removal efficiency [44]. Beyond PACs, emerging engineering controls, such as localized containment systems, offer promising approaches to mitigating airborne particle dispersion in dental clinics. By creating physical barriers around the working zone, these systems reduce particle spread before it mixes with room air, lessening reliance on room-scale ventilation [45]. Despite this potential, no CFD-based studies have comprehensively assessed the performance of localized containment systems in dental settings.

This research aims to enhance occupational safety in dental clinics by investigating the dispersion of airborne particles during the grinding or polishing of dental materials under realistic clinical conditions. The study first examines the influence of dentist working positions across four primary orientations (0°, 90°, 180°, and 270°) to analyze the relationship between workspace orientation, airflow patterns, and exposure levels to non-biological particles. Following this, the performance of portable air purifiers (PACs) with side-facing and rear-facing inlet configurations is compared to evaluate how device structure and airflow direction affect particle reduction efficiency. Additionally, the effectiveness of a localized containment system, the BC, is assessed as a mitigation strategy to minimize the spread of particles. All scenarios are simulated using CFD to replicate real-world conditions, with results validated against experimental data. By integrating advanced computational modeling with rigorous empirical validation, this study provides a comprehensive framework for understanding and mitigating particulate exposure risks in modern dental environments. The findings provide practical recommendations for optimizing dentist workspace layouts, selecting appropriate air purifier designs, and implementing evidence-based ventilation strategies, thereby contributing to safer and healthier clinical environments while supporting the continued advancement of digital technologies.

## 2. Materials and Methods

### 2.1. Physical Model and Boundary Conditions

As the first step of numerical calculation, the simulation model in this study was developed based on the actual dimensions of a dental clinic room in Thailand, as shown in Figure 1a, with the room measuring 4.34 m × 3.55 m × 2.45 m, as illustrated in Figure 1b. Minor interior elements that do not significantly affect airflow were excluded from the model. The two opposite walls are defined as Wall X and Wall Y. The key components within the room include furniture units placed against Wall X and Wall Y for storing dental instruments, and a dental chair positioned at the center of the room. During operation, the dentist sits beside the dental chair to facilitate access to the equipment attached to it. A small table is positioned nearby to hold dental components during procedures, and it serves as the main particle release point due to the use of the prosthetic grinding tool, as illustrated in Figure 1c. Moreover, the air conditioning unit is mounted on Wall Y at a height of 2.25 m from the floor, with the fan blade angle set at 45 degrees. In this study, the air conditioner is modeled to function as both an air inlet and an outlet. As illustrated in Figure 1d, the red surface represents the outlet location, while the blue surface indicates the inlet position. The dimensions of the inlet and outlet are 0.625 m × 0.135 m, displayed as blue, and 0.825 m × 0.237 m, displayed as red, respectively.

The specification of appropriate boundary conditions is a critical aspect of CFD calculations, as it directly influences the accuracy and reliability of the simulation results. Therefore, Table 1 summarizes the boundary conditions defined for both the airflow and particle simulations. The inlet is specified as a velocity inlet with a velocity of 5.5 m/s and a temperature of 24 °C, based on experimental data. The turbulence intensity at the inlet is set at 5%. The outlet is defined as an outflow boundary, serving as the exit for both air and suspended particles within the room. All wall surfaces and other physical components are modeled as no-slip walls, resulting in zero fluid velocity at the wall boundaries [46]. Additionally, the surface of the dentist is assigned a convective heat flux of 58.5 W/m^2^, following the configuration established in [47]. Moreover, in the particle calculations, particles were released from the prosthetic grinding tool, which was positioned near the dental chair, corresponding to the dentist’s actual working location. The release duration was set to 30 s, and the particles were modeled as spherical with a diameter of 0.5 μm and a density of 1300 kg/m^3^, based on the material properties of polymethyl methacrylate (PMMA) [48]. The particle release rate was set at 850 particles per second, and the initial velocity of the particles was defined as 10 m/s, based on data from high-speed rotary instruments [49,50].

**Figure 1 toxics-13-00957-f001:**
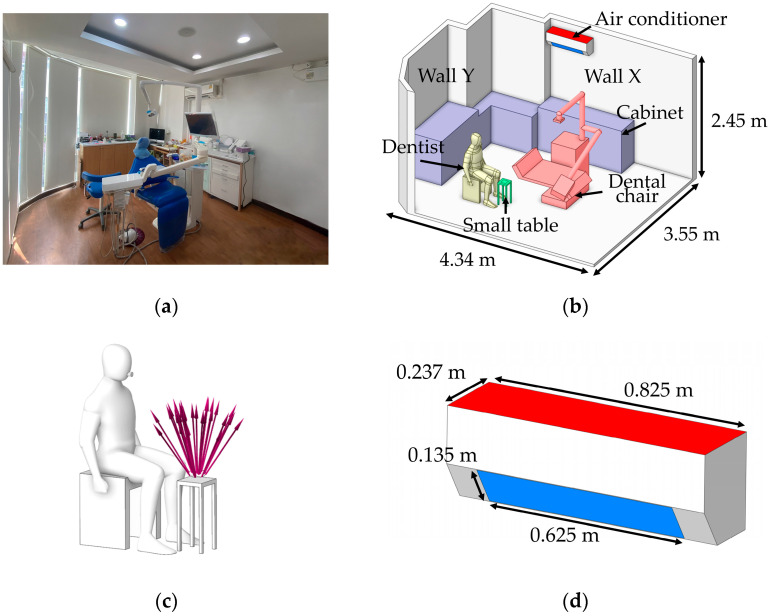
Integrated Model of Dental Operatory Architecture and Critical Boundary Definition. (**a**) The snapshot of the actual clinic in Thailand used for field validation of the CFD domain; (**b**) three-dimension CFD model illustrating operatory dimensions and key furniture placement; (**c**) definition of the particle emission source model representing PMMA grinding activity relative to the dentist’s position; and (**d**) geometric details and dimensions of the air conditioner used to define airflow inlet and outlet boundary conditions for CFD simulation.

**Table 1 toxics-13-00957-t001:** Boundary conditions of CFD simulation.

Boundary Types	Boundary Conditions
Inlet	Velocity inlet 5.5 m/s, Temperature 24 °C
Outlet	Outflow
Human Surface	No-slip wall, Heat flux = 58.5 W/m^2^
Wall and furniture	No-slip wall
Density of particle	$1300\;\mathrm{kg}/m^{3}$
Number of particles	850 pcs/s
Diameter of the particle	0.5 μm
Particle velocity	10 m/s
Particle droplet	30 s

### 2.2. Meshing

The generation of a high-quality mesh is a fundamental step in CFD simulations, playing a key role in achieving accurate and reliable results. In this study, Fluent Meshing 2022R1 was used to generate the mesh of the dental clinic model. As shown in Figure 2, the selected mesh type is poly-hexcore, a hybrid structure that combines a hexahedral core with polyhedral cells. This configuration helps reduce computation time while maintaining efficient and accurate analysis. For complex and critical areas—such as regions near the particle source, air inlets and outlets, around the air conditioning unit, and the dentist’s workspace the mesh was locally refined to capture detailed airflow and particle behavior more precisely. The mesh quality was evaluated based on the Aspect Ratio, which ranged from 1 to 15.7, falling within the acceptable standard range [46].

### 2.3. Numerical Model

#### 2.3.1. Airflow Phase Model

Based on the calculated Reynolds number, the airflow in this study falls within the turbulent flow regime. Previous research on airflow in enclosed indoor environments has also employed similar assumptions. Therefore, this study adopts the Reynolds-Averaged Navier–Stokes (RANS) approach, a common form of the Navier–Stokes equations used for analyzing turbulent flows. For the turbulence model, the RNG k-ε model was selected, which is widely used in studies of indoor airflow due to its notable accuracy, numerical stability, and computational efficiency [51,52]. Moreover, the governing equations in this model encompass the conservation of mass, momentum, energy, and the transport of turbulent kinetic energy [53].

The mass equation:(1)$$\frac{\partial \left(\rho u_{i}\right)}{\partial x_{i}}=0$$

The momentum equation:(2)$$\frac{\partial \left(\rho u_{i}u_{j}\right)}{\partial x_{j}}=\frac{\partial }{\partial x_{j}}\left[\mu_{eff}\left(\frac{\partial u_{i}}{\partial x_{j}}+\frac{\partial u_{j}}{\partial x_{i}}\right)-\frac{2}{3}\mu_{eff}\frac{\partial u_{k}}{\partial x_{k}}\right]-\frac{\partial p}{\partial x_{i}}+\rho g_{i}+S_{i}$$
(i, j, k = 1, 2, 3 and i ≠ j).

The energy equation:(3)$$\frac{\partial }{\partial x_{i}}\left(u_{i}\left(\rho e+p\right)\right)=\frac{\partial }{\partial x_{i}}\left(\lambda_{eff}\frac{\partial T}{\partial x_{i}}+u_{j}{\left(\tau_{ij}\right)}_{eff}\right)+S_{h}$$

The transport equations of the turbulent kinetic energy k and its dissipation rate ε:(4)$$\frac{\partial \left(\rho k\right)}{\partial t}+\frac{\partial \left(\rho ku_{i}\right)}{\partial x_{i}}=\frac{\partial }{\partial x_{j}}\left[\left(\mu +\frac{\mu_{i}}{\sigma_{k}}\right)\frac{\partial k}{\partial x_{j}}\right]+G-\varepsilon$$(5)$$\frac{\partial \left(\rho \varepsilon \right)}{\partial t}+\frac{\partial \left(\rho \varepsilon u_{i}\right)}{\partial x_{j}}=\frac{\partial }{\partial x_{j}}\left[\left(\mu +\frac{\mu_{t}}{\sigma_{\varepsilon }}\right)\frac{\partial \varepsilon }{\partial x_{j}}\right]+C_{\varepsilon 1}\frac{\varepsilon }{k}G-C_{\varepsilon 2}\frac{\varepsilon^{2}}{k}-C_{\mu }\eta^{3}\frac{1-\eta /\eta_{0}}{1+\beta \eta^{3}}\frac{\varepsilon^{2}}{k}$$
where(6)$$\mu_{t}=C_{\mu }\frac{k^{2}}{\varepsilon }$$(7)$$\eta ={\left(2E_{ij}\cdot E_{ij}\right)}^{1/2}\frac{k}{\varepsilon }E_{ij}=\frac{1}{2}\left(\frac{\partial u_{i}}{\partial x_{j}}+\frac{\partial u_{j}}{\partial x_{i}}\right)$$

In the k-ε equations, the parameter.

$C_{\varepsilon 1}=1.42,\;C_{\varepsilon 2}=1.68,\;C_{\mu }=0.0845,\;\sigma_{k}=0.7194,\;\sigma_{\varepsilon }=0.7194,\;\eta_{0}=4.38,\;and\;\beta_{0}=0.012$ are from Ref. [54].

#### 2.3.2. Particle Phase Model

To investigate the dispersion of airborne particles, the Lagrangian Discrete Phase Model (DPM) was employed. This model is a trajectory-based technique that tracks individual particles by treating them as a separate secondary phase, using a two-phase Eulerian–Lagrangian approach [54]. This method is widely accepted in research involving the dispersion of indoor particles. The particle trajectory is governed by the force balance equation, which determines the motion of each discrete mass point [36,55].(8)$$\frac{du_{p}}{dt}=F_{D}\left(u_{a}-u_{p}\right)+\frac{g_{x}\left(\rho_{p}-\rho_{a}\right)}{\rho_{p}}+F_{x}$$

In this simulation, pressure gradient force, Basset force, and virtual mass force were not considered, as the density ratio between air and the particles is relatively low, making the effects of these forces negligible [56,57]. Instead, the forces analyzed in this study include the thermophoretic force, Saffman lift force, and Brownian force. These forces are essential for describing the behavior of fine particles suspended in air under the influence of temperature gradients, shear flow, and random motion. They have a direct impact on the dispersion and accumulation of particles in high-risk zones [54].

The CFD simulation was carried out using ANSYS Fluent 2022 R1, a widely used software for analyzing airflow in indoor and enclosed environments [58]. The total simulation time was set to 3 min to track particle behavior across three-time intervals: 0–60 s, 61–120 s, and 120–180 s. The coupling between pressure and velocity fields was handled using the SIMPLEC algorithm, which offers faster convergence. Spatial discretization for all variables, including momentum, energy, turbulent kinetic energy, and turbulence dissipation rate, was performed using the Second-order Upwind Scheme. For pressure, the PRESTO! The scheme was selected as suitable for simulations involving rapid changes in velocity and pressure, especially in confined spaces, and is commonly used in particle dispersion studies [59]. For temporal discretization, the Second-order Implicit Scheme was employed due to its advantages in stability and accuracy when tracking time-dependent changes in the flow field.

In this transient simulation, a fixed physical time step of Δt = 0.005 s was applied. Although no explicit Courant–Friedrichs–Lewy (CFL) limit was imposed, solver monitors indicated values of order unity, with localized peaks near the jet-refined region reaching CFL max = 2.75, while the domain-mean CFL remained below 1 throughout most of the computation. The implicit, pressure-based formulation with bounded second-order upwinding maintained stable convergence. Each time step was iterated until velocity and continuity residuals < 10^−4^ and energy residuals < 10^−6^; the monitored velocity and pressure histories showed no oscillations, and the global mass imbalance was <1%, confirming numerical stability.

### 2.4. Experimental Method

#### 2.4.1. Measurement Devices

Field data collection was conducted to support the simulation results. The research team measured two physical parameters air velocity and particle concentration —within the dental clinic. Table 2 presents the specifications of the measuring instruments. For air velocity measurement, the TESTO anemometer was selected, capable of measuring wind speeds up to 20 m/s with an accuracy of ±5 m/s. To measure particle concentration, a portable laser particle counter was used, which can detect particles of three sizes: 0.3, 0.5, and 5 μm. Although the measurement accuracy is ±10%, the device is considered appropriate for the scope of this study.

#### 2.4.2. Field Measurement

##### Air Velocity Measurement

In this study, wind speed measurements were conducted at seven locations, labeled P1–P7 as shown in Figure 3. Point P1 represents the outlet of the air conditioning unit and was also used to define the initial velocity boundary condition in the simulation. Points P2–P7 were used to validate the accuracy of the simulation model by comparing measured values with simulation results. These points were arranged at horizontal intervals of 0.5 m and vertical drops of 0.4 m per point, aligning with the approximate 45° air discharge angle of the fan blades.

Each measurement point was carefully evaluated and measured five times, with the average values used for further analysis. As shown in Table 3, the wind speed was observed to decrease with increasing distance from the air conditioner outlet, which corresponds with the expected velocity behavior over greater distances.

##### Quantifications of Particle Concentrations

Particle concentration measurements were also conducted during the grinding or polishing of dental devices using a particle counter to assess the level of particle accumulation at the target locations, as shown in Figure 4. The measurement process was divided into two stages.

Particle Measurement at the Source: This step aimed to evaluate the quantity of particles generated during the dental procedure and to define the particle release boundary condition in the simulation. To minimize interference with the dentist’s workflow, the measuring device was placed as close as possible to the working area at a height of 0.45 m from the floor, corresponding to the typical operating height. Data collection was conducted over 30 s, based on the average time required to polish a single dental prosthesis by an experienced practitioner [60]. This duration also corresponds to the period during which the highest concentration of particles is typically released in dental procedures [61]. As shown in Table 4, the particle count measured at the source reached 25,000 particles.Particle Concentration Measurement at Various Locations in the Room: To assess the particle concentration within the room, five measurement points were designated as S1–S5, which included the four corners of the room (S1, S2, S3, S4) and the other at the center of the room (S5), as shown in Figure 4c. These positions were selected because they are likely to generate recirculation and vortex zones, which can directly affect particle transport and result in significant accumulation [62]. Data collected from these positions were used to compare and validate the simulation results on particle concentration. During the experiment, the particle counter was placed 15 cm away from both adjacent walls and at a height of 0.45 m, consistent with the release height at the particle source. Each position was measured five times, with each measurement lasting 1 min. Owing to the limited number of particle counters, measurements were performed in a systematic sequence, with the clinic room undergoing basic cleaning for 45–60 min after each round to restore a safe baseline environment before the next measurement session. As shown in Table 4, the highest particle concentration was observed at S1, followed by S4, then S2, S3, and S5, in that order. These findings support the hypothesis that the corners of the room are prone to the formation of recirculation and vortex zones, resulting in higher particle accumulation compared to the center of the room. Additionally, the results help identify potential high-risk areas within the clinic, particularly at S1 and S4, where particles tend to accumulate or be carried by airflow in significant quantities.

Furthermore, experimental data also revealed that the majority of detected particles were 0.3 and 0.5 µm, which dominated the measured particle counts during PMMA grinding and are capable of penetrating deep into the lungs, indicating a potential respiratory risk. Consistent with previous studies that commonly adopt 0.5 µm as a representative submicron class for indoor aerosol transport and exposure assessment [63,64], this size was selected as the primary case for CFD–experiment comparison due to its prevalence in the measured data and its relevance to respirable deposition.

**Table 4 toxics-13-00957-t004:** Quantitative particle concentration values for CFD model validation.

Positions	Starting Point at Particle Source (30 s)	S1 (60 s)	S2 (60 s)	S4 (60 s)	S4 (60 s)	S5 (60 s)
Particle concentration $(\mathrm{kg}/m^{3})$	25,000 Particles	$3.73\;\times {\;10}^{-10}$	$2.43\;\times \;10^{-10}$	$1.94\;\times {\;10}^{-10}$	$3.06\;\times \;10^{-10}$	$2.10\;\times \;10^{-10}$

**Figure 4 toxics-13-00957-f004:**
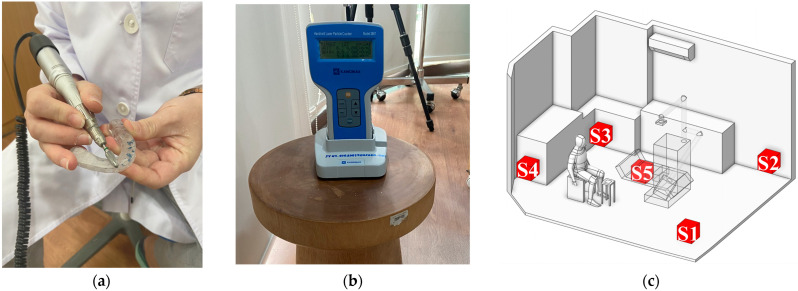
Experimental setup for real-time particle monitoring and source validation. (**a**) Close-up of the simulated high-emission procedure: mechanical grinding of PMMA (Dental hard splint); (**b**) The handheld particle counter used for real-time measurement of particulate matter concentration to validate CFD results; (**c**) Schematic showing reference cube and camera positions aligned with the physical locations of five particle sampling cubes (S1 to S5) for experimental measurement.

### 2.5. Local Concentration Reduction

#### 2.5.1. Working Zone Location

To evaluate the risk of particle exposure, the green box in Figure 5 defines a localized region encompassing the dentist’s working and breathing zones. With dimensions of 0.876 m × 0.775 m × 0.732 m, this region serves as a monitoring volume where particle ingress is recorded to facilitate targeted prevention and control strategies.

#### 2.5.2. Impact of Portable Air Cleaner Intake Geometry on Mitigation

Currently, a wide range of air purifiers is commercially available, each tailored to specific operating conditions. Accordingly, this study assessed the performance of two distinct models in reducing particle concentration. As shown in Figure 6, the models differ in inlet configuration: (a) side air inlet and (b) rear air inlet. Both were designed with identical inlet and outlet areas of 0.275 m^2^ and 0.858 m^2^, respectively. The inlet velocity was specified as 2.5 m/s, while the outlet boundary was defined as outflow.

Figure 7 illustrates the placement of the air purifier, positioned either near the particle emission source or adjacent to the working area. These locations, identified in previous studies [40,41,42,43], have been recognized as the most effective points for installing air purifiers.

#### 2.5.3. Box Dust Collector

The BC model was developed based on the prototype employed in the clinical experiment. As illustrated in Figure 8, it has dimensions of 0.36 m × 0.28 m × 0.24 m and features hand insertion openings with a diameter of 0.11 m. The device was placed on a small working table, where particle generation was confined within the boundary of the BC, thereby avoiding particle diffusion in the dental clinic [65].

## 3. Results and Discussions

### 3.1. Validation

To ensure the reliability of the simulation results, this study conducted a validation process by comparing the simulation outcomes with experimental data. The validation was divided into two cases for air velocity variation and particle concentration.

#### 3.1.1. Airflow Velocity and Grid Sensitivity

The validation of air velocity results was conducted in conjunction with a mesh independence test, which is crucial for ensuring both computational accuracy and efficiency of the simulation. Selecting an appropriate mesh size helps optimize both computational resources and simulation time. In this study, three mesh sizes were generated: 1,824,351 nodes (1.10 million elements), 3,132,859 nodes (1.88 million elements), and 6,372,269 nodes (3.82 million elements). The results from each mesh were compared against experimental data, as illustrated in Figure 9. The coarsest mesh, with 1.8 million nodes, exhibited the highest maximum error at 31.82%, indicating an insufficient spatial resolution for capturing complex flow phenomena near dental room equipment and room boundaries. The intermediate mesh size (3.1 million nodes) demonstrated a substantial improvement, significantly reducing the maximum error to 9.10%. While the finest mesh with 6.3 million nodes provided slightly higher accuracy, the maximum error was only marginally lower at 8.81%, despite more than doubling the computational cost. Considering the trade-off between accuracy and computational demand, the mesh with 3,132,859 nodes (1.88 million elements) was selected as the most suitable configuration. It provided results within an acceptable range of accuracy while also efficiently reducing computational time and resource usage. The maximum air velocity error of 9.10% is considered acceptable based on standard validation criteria at a 10% confidence level [66]. Therefore, the results show good agreement, with consistent trends observed between the simulation and the experiment.

#### 3.1.2. Particle Concentrations Validation

For the validation of particle dispersion, the simulation included the creation of designated monitoring zones that correspond to the field measurement positions, as illustrated in Figure 4c. Each monitoring zone measured 0.3 m × 0.3 m × 0.7 m, matching the suction rate of the particle counter. The volume summary method was used to calculate particle accumulation at each location over a 1 min period, and the results were compared with field measurement data. This comparison allowed for the evaluation of the accuracy and reliability of the simulation outcomes in this study.

Figure 10 presents a comparison of particle concentration between the simulation and field measurements at positions S1, S2, S3, S4, and S5, with percentage errors of 6.5%, 9.6%, 5.4%, 7.0%, and 8.5%, respectively. These discrepancies are likely due to several limitations, including constraints of the measurement device, the complexity of the experimental procedure, environmental fluctuations that could not be fully controlled, and potential human error during data collection. All these factors may contribute to deviations between the simulated and experimental results. Considering these factors, the maximum observed error of 9.6% is still within an acceptable and reliable range for validating the particle dispersion simulation conducted in this study.

### 3.2. Investigation of Health Risks Location Within the Clinic

The risk assessment in the clinic was conducted in two steps: (1) analysis of the airflow field to examine movement patterns influencing particle dispersion, and (2) evaluation of particle distributions to characterize the spread of particles within the room.

#### 3.2.1. Airflow Field

The airflow patterns in the room were visualized using streamlines, with numerical labels denoting each sequential step of the flow. In line with the study objectives, the results were presented at two height levels: (1) 0.45 m, representing the particle generation zone, and (2) 1.1 m, corresponding to the dentist’s breathing height.

At a height of 0.45 m, as shown in Figure 11a, sorting by number, the air released from the air conditioner flows according to the numbers 1, 2, 3, 4, 5, and 6. Most of the air released from the inlet flows directly toward the working zone. At number 2, airflow carries particles generated during dental material grinding and disperses them throughout the room, as shown in number 4. The airflow then impinges on the wall, resulting in a reduction in velocity reduction and separation into two streams. These separated flows subsequently collide with the room corners, forming recirculation zones, most prominently at positions S1 and S4. At these locations, repeated vortex-like circulations occur, which reduce airflow velocity and facilitate the accumulation of particles. By contrast, position S5, located at the center along the initial airflow path, demonstrates better ventilation than the other positions.At a height of 1.1 m, as shown in Figure 11b, sort by number, 1, 2, 3, 4, 5, 6, 7, 8, and 9. The airflow predominantly forms recirculating currents originating from the air conditioner outlet and extending into all corners of the room. At numbers 7, 8, and 9, the flow repeatedly returns toward the center of the room, reflecting inefficient ventilation. This pattern indicates that particles released into the air are likely to recirculate to their original positions, thereby increasing the dentist’s risk of exposure and inhalation.

#### 3.2.2. Particle Distribution

According to Figure 12, during the initial stage (1–5 s), most particles remained concentrated near the source, close to the dental workstation. A distinct upward dispersion was observed, driven by the source momentum and the influence of overhead airflow passing above the dentist, with some particles beginning to deviate along the primary airflow direction. Between 10 and 30 s, particles became more widely distributed, with noticeable horizontal transport from the center toward the side walls, particularly in the upper-left and upper-right corners, where accumulation increased. This effect was attributed to the formation of recirculation zones, which act as pocket regions that trap air and hinder dispersion. During 60–90 s, the dispersion process entered a quasi-saturation stage, with particle concentrations peaking in several areas. Settling and repeated recirculation occurred, but without efficient removal from the space. Although some particles drifted toward the outlet, the room geometry and equipment layout induced persistent recirculating flows, especially in the central area. By 120–180 s, particle levels declined in certain regions, particularly near the air outlet. However, poorly ventilated zones, such as the corners and the space beneath the table, retained particles, underscoring the limitations of the existing ventilation system in removing airborne contaminants within the 3 min timeframe.

### 3.3. Spatial Concentration Reduction Measures

#### 3.3.1. Optimal Working Position Assessment

This approach characterizes particle dispersion patterns corresponding to different working positions, categorized into four orientations: 0°, 90°, 180°, and 270°, as illustrated in Figure 13a–d. These orientations represent the dentist’s location in each case study. The objective of this analysis was to determine the position associated with the lowest particle exposure, thereby minimizing the dentist’s risk of inhalation. Analysis of the accumulated particle concentration within the designated working zone (as defined in Figure 5 and detailed in Figure 14) reveals a dramatic dependence on the operator’s position relative to the primary ventilation flow. The highest risk positions (180° and 90°) placed the operator against the primary airflow or beneath the air conditioning unit (in low velocity zones), significantly facilitated particle accumulation. The 180° orientation resulted in the highest peak concentration at 1.25 × 10^−9^ kg/m^3^, followed closely by the 90° position at 6.50 × 10^−10^ kg/m^3^. The optimal positions (0° and 270°), conversely, aligned with or leveraged the general ventilation stream showed significantly lower accumulated concentrations. The 0° position exhibited the lowest peak concentration at 3.81× 10^−10^ kg/m^3^, and the 270° position at 4.10 × 10^−10^ kg/m^3^. This data definitively links localized air speed and direction to the accumulation of toxicants. The stark contrast in particle accumulation is directly attributable to the mechanism of airflow interference. Our CFD-validated results show that the 180° working orientation, which places the operator directly against the primary ventilation flow, not only creates significant stagnation zones but also amplifies the operator’s breathing-zone concentration by approximately 65% compared to the optimized 0° position. In the high-risk 180° arrangement, the dentist’s body acts as a macroscopic physical obstruction, transforming the laminar flow into turbulent localized recirculation loops. These loops effectively trap airborne particulates, leading to pronounced stagnation zones and the highest measured concentrations directly in the breathing zone, posing the greatest inhalation risk. The lower accumulated concentrations at 0° and 270° confirm that alignment with the general ventilation stream minimizes this local flow impedance, allowing for more efficient particle transport away from the operator.

The temporal analysis of particle clearance highlights a critical gap in current safety protocols. During the initial 0–40 s particle release period, all rotated positions exhibited high, potentially acute exposure levels. While general ventilation eventually reduced particle concentrations across all positions, the values did not converge to comparable, safer levels until approximately 160 s. The relatively long duration (approximately 160 s) required for environment reduction before reaching safe conditions is clinically unacceptable, particularly in high-volume settings. It confirms that reliance solely on general ventilation leaves both the dental professional and the subsequent patient vulnerable to acute, high-dose exposure during and immediately following a procedure. Therefore, the 0° position was selected for further investigation. This selection is based on its status as the minimum-vulnerability baseline, providing the most favorable fluid dynamic environment. However, the requirement for additional control measures, even in this optimized position, underscores the necessity of moving beyond simple positional changes toward proactive, airflow-aware engineering controls to ensure immediate and sustained safety. Our findings advocate for a fundamental redesign of operatory layouts, incorporating principles of environmental fluid dynamics to minimize professional exposure risks.

#### 3.3.2. Effects of Air Purifiers on Particle Mitigation Improvement

The effectiveness of both air purifier models in reducing particle accumulation was evaluated in two cases: (1) the whole room and (2) the localized working zone, or the designed region. This analysis facilitated the identification of the most suitable model for the targeted application. The dentist’s working position was fixed at the initial orientation of 0°. Figure 15 presents the particle accumulation curves for three scenarios: the baseline case without protective measures, and the cases employing Air Purifier Model 1 and Model 2.

From Figure 15a, during 0–30 s, particle concentration increased markedly, corresponding to the particle release phase. In the baseline case, the maximum concentration reached 5.00 × 10^−10^ kg/m^3^. After the release ended, concentrations in all three cases gradually declined to safe levels. Compared with the scenarios employing the air purifiers, the devices reduced overall particle concentration by up to 40%, with a maximum of 3.20 × 10^−10^ kg/m^3^. Although both models demonstrated effective average concentration reduction, Model 2 outperformed Model 1, achieving approximately 5% greater reduction in average particle concentration.

The assessment of the designed region highlights the need to select air purifiers according to specific operating conditions. As shown in Figure 15b, Air Purifier Model 2 reduced particle concentration by up to 80% compared with the baseline case, achieving a peak concentration of 7.03 × 10^−11^ kg/m^3^. By contrast, during the same period, Model 1 achieved only a minor reduction in the number of cases. This performance gap underscores the importance of outlet design for localized applications. Following the particle release phase, Model 1 showed limited capability in further reducing concentration, whereas Model 2 consistently maintained low levels. The superior performance of Model 2 is attributed to the size and placement of its outlet, which facilitates direct capture of airflow and particles transported from the working zone. In contrast, the side outlet of Model 1 was less effective in capturing particles from the emission source. Therefore, in environments with sustained particle generation, careful consideration of outlet orientation and placement is essential to maximize removal efficiency.

To maintain exposure control during high-emission tasks, portable air cleaners should be sized by ${CADR}_{req}\approx {ACH}_{target}\times V$. For our single-chair operatory (~38 m^3^), a minimum of 5–6 ACH implies a flow rate of ~190–230 m^3^/h. A 20–30% safety margin is recommended to offset filter-loading and placement losses; alternatively, multiple units may be used with additive CADR under well-mixed conditions. Because CADR depends on the filter media and their loading state, selection should consider both CADR and submicron CADR (sCADR), as well as the expected decay of CADR with use [67].

Generalizability, these results were obtained in a single-chair operatory (38 m^3^) with a unidirectional supply stream. The mechanistic effect of a rear-inlet PAC aligned with the primary supply should be transferable to similar small rooms where a dominant room jet exists. However, absolute reductions will vary with ACH (i.e., CADR per room volume), ceiling height, and diffuser/return layout; thus, percentages should be re-estimated for dissimilar geometries, while the relative advantage of the rear-inlet configuration is expected to persist under comparable flow topologies.

#### 3.3.3. Effectiveness Zones of Preventive Measures

To evaluate location-specific risks within the room, the space was divided into nine zones (1–9). Zones 1, 2, 4, and 5, representing the dentist’s working areas, were given special consideration. For each zone, the efficiency of both air purifier models in reducing particle concentration was assessed, enabling the identification of high- and low-risk areas.

From Figure 16a, during the initial dispersion stage (0–60 s), Model 1 (left panel) exhibited inconsistent performance; for example, Zone 1 showed a negative efficiency (−3.92%), reflecting uneven particle accumulation and dispersion. In contrast, Model 2 demonstrated stable performance across multiple zones, with efficiencies ranging from 30% to 42% and no negative values observed, highlighting its superior capability in ventilation and particle control during the peak dispersion phase.

From Figure 16b, the overall efficiency across the 180 s period confirmed that Model 2 consistently outperformed Model 1, particularly in the dentist’s working zones. Model 2 also exhibited a more uniform distribution of efficiency. These findings indicate that Model 2 is more suitable than Model 1 for environments characterized by high particle generation over short durations.

**Figure 16 toxics-13-00957-f016:**
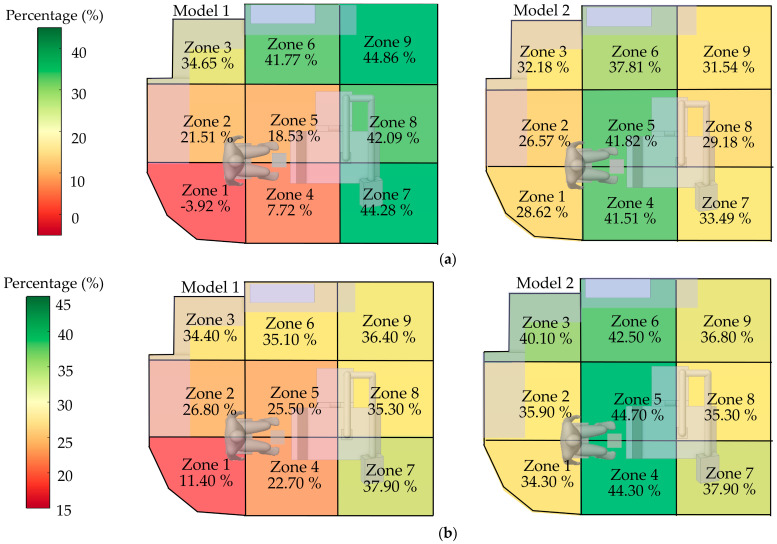
Localized mitigation efficacy (percentage reduction) of PAC Models across defined operatory zones: (**a**) the critical initial release phase (0–60 s), and (**b**) over the full clearance period (0–180 s).

#### 3.3.4. Investigation of Particle Mitigation Effects Using Box Dust Collector

This analysis explicitly investigates the efficacy of the BC as a localized source containment measure, implemented to prevent immediate particle dispersion into the designed working and breathing zones. Source containment is a fundamentally sound engineering strategy, and our results confirmed its transient efficacy [65,68]. As demonstrated in Figure 17a, the BC significantly restricted particle dispersion during the initial phase (0–60 s) of generation. During this critical period, the majority of released non-biological particles were successfully retained inside the containment volume, thereby delaying contamination of the surrounding operatory areas. This finding validates the principles of utilizing a localized containment device to provide essential initial inertia against the immediate development of the aerosol plume, which is particularly relevant for short-burst, high-emission clinical tasks. However, the analysis of prolonged operation revealed a critical limitation inherent to this localized containment design. Despite its effective control of early-stage particles, with a maximum concentration in the designed region limited to 3.22 × 10^−10^ kg/m^3^ during the initial phase, the system’s inability to maintain a perfect aerodynamic or physical seal meant that continuous particle generation gradually compromised the containment integrity. Over time, particles inevitably found leakage pathways, dispersing into the room and the defined working zone, as visually supported by the development of the particle cloud shown in Figure 17b. This accumulation of leaked particles, evidenced by the rising concentration over the extended period, compromises the long-term occupational safety of the system when deployed in isolation.

This transient failure highlights a fundamental engineering challenge: while a BC excels at initial mass constraint, it often falls short in maintaining sustained clearance kinetics due to inevitable gaps, minor leaks, and user handling. To minimize the dentist’s exposure effectively and sustainably, reliance on the BC alone is insufficient. Therefore, the data leads us to the critical conclusion that source containment must be synergistically paired with continuous air filtration. The BC is therefore recommended in combination with the Air Purifier Model 2.

#### 3.3.5. Interoperability Between PACs and BC

The integrated use of PACs (Model 2) with a BC aims to enhance the reduction in particle concentrations in the dentist’s working zone by combining source containment with continuous filtration within the working and breathing zones. This approach is intended to overcome the limitations of deploying either device in isolation.

Figure 18 shows the temporal evolution of particle concentration in four cases within the dentist’s working zone (designed region), as defined in Figure 5. The combined use of PACs and BC yields the lowest and most stable levels throughout the 0–180 s period, with a peak concentration of only 4.67 × 10^−11^ kg/m^3^. Using PACs alone produces a slightly higher peak of 7.03 × 10^−11^ kg/m^3^, whereas BC alone delays early dispersion; however, once saturation is reached, particles escape, and the peak rises to 3.22 × 10^−10^ kg/m^3^. Comparing across cases, the lowest values in the PACs + BC configuration reflect a synergistic mechanism: the BC constrains source emissions in the initial phase. In contrast, the PACs continuously remove escaped particles along the main flow direction. Consequently, this synergistic effect demonstrates the multiplicative benefits of combined mitigation strategies, exceeding the individual effectiveness of either technology alone.

Our investigation confirms a fundamental dichotomy in dental workplace safety. The highest levels of particulate control demand engineering solutions that often conflict with real-world spatial and ergonomic constraints. In compact operatories, deployment of combined PAC + BC can be limited by spatial footprint/clearance, ergonomic interference with clinician–assistant–patient workflows, acoustic/draft comfort, power/cable management, and cost/maintenance (e.g., filter replacement and CADR decay under loading). While simple containment devices are prevalent, achieving robust, long-term safety requires a sophisticated, multi-layered strategy. These simulation results decisively establish the superiority of a hybrid synergistic mitigation strategy for controlling particle dispersion in dental environments. The rear-inlet portable air cleaners (PACs) effectively reduced overall room-air particle concentrations, making them particularly advantageous in clinics with limited ventilation capacity systems. Nevertheless, for precise and comprehensive control, especially to minimize the risk of acute toxic exposure within the dentist’s breathing zone, we recommend combining PACs with a BC, which contains particles at the source and reduces their ingress into the working area. The joint deployment of these two measures yielded a profound synergistic effect, demonstrating that source confinement (BC) and continuous filtration (PACs) function as a robust two-stage defense. Design attributes that alleviate the above barriers include a compact, mobile form factor, low-noise operation, and an adjustable inlet orientation to align with the primary room jet. This approach is absolutely recommended as the best method for long-term occupational safety and minimizing inhalation risks for dental professionals and their patients. While such constraints may condition absolute reductions, they do not alter the relative advantage observed for the rear-inlet configuration in this study. A concise summary of the relative mitigation efficiencies across the tested configurations is presented in Table 5, where the combined PACs and BC system demonstrated the most outstanding performance stability and exposure reduction, particularly under high-emission dental tasks. These findings confirm that a coordinated, two-tiered strategy—combining source confinement with continuous air purification—provides the most effective and sustainable protection in in-house laboratory–clinic settings.

**Figure 18 toxics-13-00957-f018:**
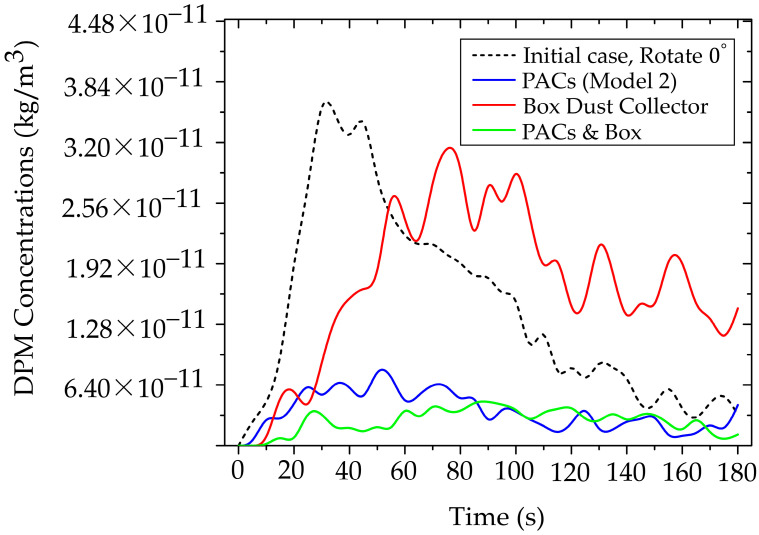
Synergistic mitigation efficacy of combined engineering controls.

### 3.4. Mandating a New Paradigm for Airborne Toxicant Control in Digital Dentistry

This comprehensive and integrated study necessitates a fundamental shift in the paradigm of occupational safety. We focus on the emerging hazards of non-biological particulates originating from advanced dental materials, which are equally critical to mitigate. A critical finding has been identified that significantly elevates the risk profile of modern dental practices. Our data rigorously demonstrated that airborne synthetic materials introduce an equivalent toxic burden, compelling the immediate adoption of proactive, engineering-based controls. This redefinition of clinical standards is essential for safeguarding the health and ensuring the long-term safety of both personnel and their patients. The unrecognized toxicological threat is the rapid expansion of digital fabrication technologies (3D printing, CAD/CAM milling) [69]. Introduces an array of particulate emission sources, including PMMA, resin-based composites, zirconia, and polymers directly into the confined operatory unit. The hazard posed by synthetic particulates is not limited to in-house laboratory settings; it spans the entire scope of clinical care. Intraoperative material adjustments from minor surgery in the dental clinic (e.g., trimming surgical guides during implant operation) to advanced procedures requiring a major operating room (e.g., orthognathic surgery or complex implant reconstruction) introduce these particulates into low-ventilation or semi-sterile fields. In all these environments, both clinical personnel and patients are exposed to the potential inhalation of suspended synthetic particulates. This exposure constitutes a critical hazard, potentially leading to respiratory irritation, inflammatory responses, or chronic accumulation of synthetic particulates in the lungs of the patients present during the procedure [69,70]. We emphasize that effective airborne material control must be codified as a core component of operational safety, extending general occupational hygiene to encompass total patient care across all procedural settings.

### 3.5. Integrating Toxicology and Dental Safety Narratives

This analysis directly translates our engineering results into specific health risk narratives within the dental context. This determination is strongly supported by the extensive body of prior research documentation among dental professionals [20,21,22,23,24,29,30]. Based on the immediate systemic absorption of synthetic compounds and the possibility of acute airway inflammatory responses in the dental operator due to inhalation of PMMA and resin microspheres, we determine that the highest peak concentrations observed at the 180° orientation (Figure 14) cannot be attributed to a simple airflow anomaly; rather, it represents a profound, quantifiable failure of ergonomic design in preventing exposure. This inherent risk is particularly pronounced during chairside material adjustment of dental devices. Furthermore, our CFD visualization of low-velocity areas (corner recirculation and stagnation zones) allows us to precisely identify toxicant reservoirs, emphasizing the chronic inhalation risk posed by the sustained presence of respirable PM2.5 particles in these non-ventilated zones, which can contribute to documented pulmonary complications among dental professionals. Ultimately, the proven efficacy of the hybrid PACs + BC system provides effective acute toxic dose reduction (by reducing peak concentrations) for both the dentist and the patient present during high-emission procedures (e.g., grinding surgical guides or dental splint, etc.). This strategic control is visually summarized in Table 5, which emphasizes the significant reduction in peak concentration and airborne residence time. This control is crucial for mitigating the long-term pulmonary burden associated with inhaling synthetic polymer dust.

**Table 5 toxics-13-00957-t005:** Summary of mitigation efficiency for particle dispersion control devices.

Mitigation Strategy	Mechanism of Action	Peak Particle Concentration Reduction	Breathing-Zone Concentration Reduction	Advantages	Limitations	Recommended Clinical Application
Baseline (No Control)	Natural room airflow only	—	—	Serves as control for CFD validation	High particle accumulation; prolonged airborne residence	Used for comparison and reference
Portable Air Cleaner (PACs Model 2)	Continuous air suction aligned with primary airflow direction	Up to 80% reduction	Up to 65% reduction	Effective at whole-room and local scales; portable and easy to install	Requires optimal placement; airflow alignment critical	General dental procedures, polishing, and prolonged tasks
Dust Box Collector (BC)	Localized source containment via frontal capture zone	Approx. 40–60 s delay in plume development	35–45% transient reduction	Effective for early-stage containment; minimizes initial spread	Limited by particle leakage; efficiency decreases over time	High-emission short-duration tasks (e.g., grinding, trimming)
PACs Model 2 + BC	Synergistic combination of local containment and continuous extraction	Up to 85% total reduction in peak level	Sustained 75–80% reduction over 180 s	Strongest and most stable control; reduces residence time and exposure	Requires coordination of airflow directions and more spaces	Optimal for high-risk scenarios (3D-printing adjustments, chairside material grinding)

### 3.6. Translating Risk into Control: Ergonomics, Material-Specific Considerations, and Future Research Mandates

Our engineering analysis confirms the limited efficacy of single-point mitigation, decisively establishing the synergistic superiority of the hybrid mitigation system: the PACs for ambient filtration combined with a BC for source confinement. While scientifically validated, the implementation of solutions that require external power hoods faces a critical spatial-ergonomic conflict rooted in fundamental differences between operational environments. Dental clinics, unlike dedicated dental laboratories, are primarily designed for patient throughput and direct clinical care, rather than laboratory processing. The architectural and functional constraints of a clinical operatory—proximity to the dental unit, and the requirement for operator mobility—make the installation of complex fixed infrastructure virtually impossible. We recommend exploring the feasibility of internal augmentation within existing BC units by integrating specialized adsorbent or capture media (e.g., filter materials). It is theorized to significantly reduce Dispersed Particulate Matter (DPM) concentrations directly at the source, effectively mitigating the risk of inevitable leakage. The limitation of simulating only fine particles (0.5 μm from PMMA) highlights a critical knowledge gap that necessitates immediate research to inform future safety guidelines.

Multi-material and longevity studies: Future research must characterize particle profiles with distinct aerodynamic sizes, densities, and surface properties for the full spectrum of modern dental materials, including emerging 3D-printing resin-based and hybrid materials [71]. Crucially, studies must quantify the operational longevity (i.e., saturation time) of internal capture media for specific composites to prevent catastrophic particle breakthrough and ensure sustained occupational safety.Additive manufacturing trends: Future research should account for the unique challenges of 3D printing, specifically analyzing the potential for the release of volatile organic compounds (VOCs) during post-processing and exploring mitigation approaches [72].Refinement of CFD models: to maximize predictive power, future CFD models must integrate refined human factors (dentist/patient breathing patterns, thermal effects, and continuous room usage) and evaluate the sustained efficacy of control systems (PACs, BC) under diverse, continuous procedural conditions.The emission source was modeled as a steady surrogate with fixed release rate and initial jet velocity, and no formal sensitivity analysis was performed. Consequently, clinically relevant variability (tool type/speed, material removal rate, operator technique), transient bursts/duty cycles, and polydisperse size distributions were not parameterized. This simplification may bias estimates of short-term concentration peaks, plume intermittency, and local hot-spot magnitudes relative to conditions with time-resolved, case-specific emissions.Validation readings were acquired in quintuplicate (*n* = 5) and reported as mean ± SD. The handheld particle counter used for 0.3–5 µm channels has a stated accuracy of ±10% stated accuracy (2.83 L·min^−1^), and the hot-wire anemometer for air velocity has a stated accuracy of ±5%. Additional contributions arise from probe positioning/angle sensitivity in strong gradients as well as counting-statistics and coincidence at elevated concentrations. Replicate SDs were modest and remained within these specifications; consequently, the maximum CFD–measurement deviation (9.6%) falls within the instrument-limited uncertainty envelope and is not indicative of systematic model bias.Future work will extend to multiple dental materials (e.g., PMMA, resin composites, zirconia) with size-resolved, polydisperse emission spectra, and will assess long-term cumulative exposure by coupling transient CFD with time-activity schedules and maintained CADR/air-cleaning performance. Exposure will be summarized using shift-level occupational metrics (e.g., 8-h TWA, cumulative inhaled dose) to link engineering controls to health-relevant endpoints.

Beyond immediate applications, the findings advocate for a paradigm shift in dental occupational safety from reactive measures toward proactive engineering-based prevention. Future clinical environments should integrate localized exhaust systems, automated filtration units, and intelligent air-quality monitoring, synchronized with procedural phases. As digital dentistry and in-house fabrication continue to expand, guidelines must evolve to incorporate non-biological particulate exposure standards, aligning with those already established for industrial polymer and composite materials.

## 4. Conclusions

This integrated computational and experimental study decisively establishes a new, evidence-based paradigm for airborne exposure control in the contemporary dental operatory, compelling a shift from reactive hygiene to proactive engineering management. We conclude that the management of non-biological particle exposure is fundamentally determined by two interrelated factors: airflow management and synergistic engineering control. Optimizing the dentist’s working orientation to align with the dominant airflow (0°) is the most immediate, no-cost intervention, proven to reduce breathing-zone concentrations by up to 65%, thus confirming the necessity of airflow-aware ergonomics in clinical workflow design. While individual devices offer localized benefits, the integrated deployment of the rear-inlet portable air cleaner (PACs, Model 2) with the BC produced the most effective and sustained reduction across the whole 180 s procedure, making this synergistic combination the most appropriate best practice for short, high-emission tasks. Adoption in practice ultimately depends on balancing CADR and placement with footprint, noise, ergonomics, and serviceability, which may condition absolute gains without altering the relative efficacy ranking. This research represents a novel interdisciplinary advancement, successfully integrating validated CFD modeling with occupational toxicology to provide actionable safety solutions and provides a robust foundation for future occupational exposure standards that must evolve to incorporate non-biological particulate exposure standards, ultimately ensuring a safer and compliant clinical environment for staff and patients in the digital dentistry era.

## Figures and Tables

**Figure 2 toxics-13-00957-f002:**
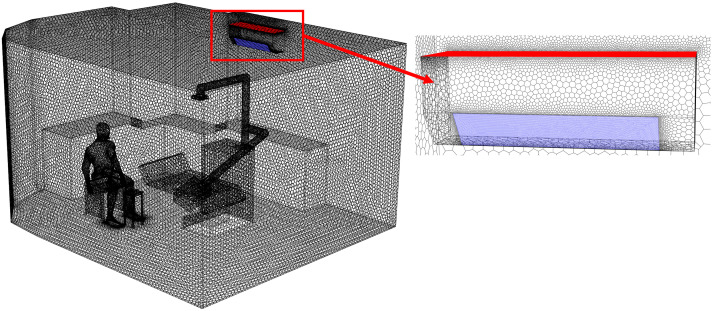
High-fidelity poly-hexcore mesh with localized refinement for CFD simulation.

**Figure 3 toxics-13-00957-f003:**
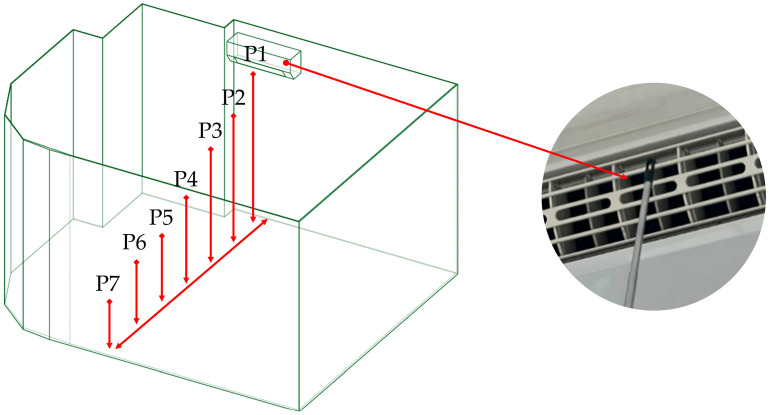
Air velocity measurement localizations for CFD validation.

**Figure 5 toxics-13-00957-f005:**
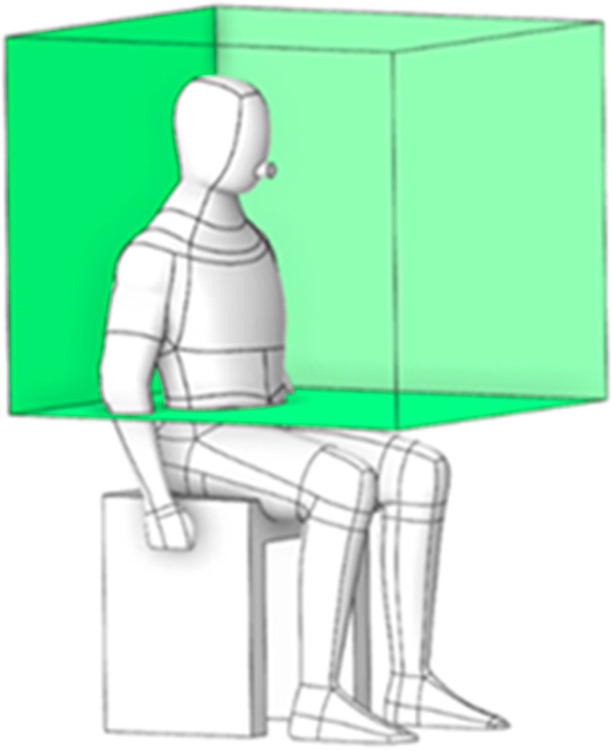
Definition of the quantified working and breathing zone.

**Figure 6 toxics-13-00957-f006:**
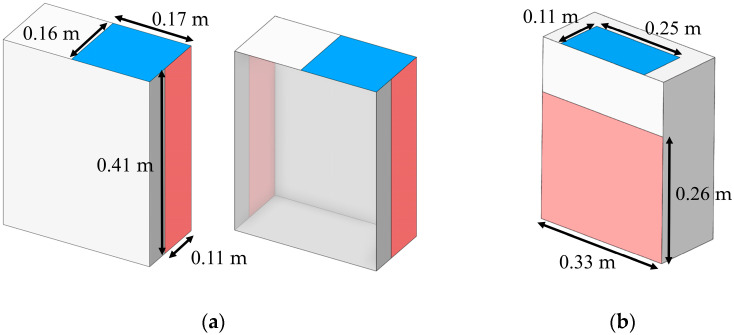
Geometric configurations and dimensions of portable air cleaners (PACs) modeled (**a**) PACs Model 1; (**b**) PACs Model 2.

**Figure 7 toxics-13-00957-f007:**
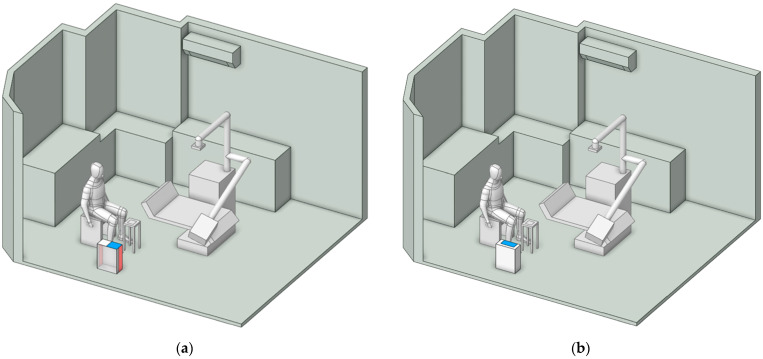
The positions of the air purifier for mitigation analysis (**a**) PACs Model 1; (**b**) PACs Model 2.

**Figure 8 toxics-13-00957-f008:**
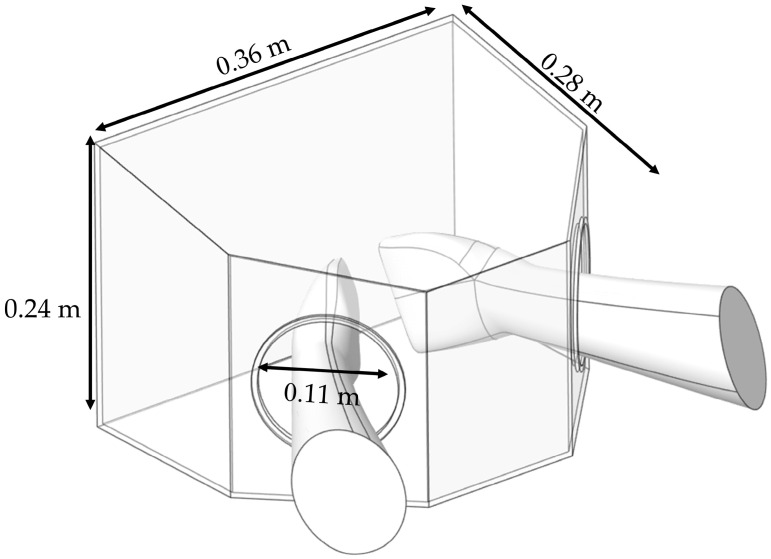
Geometric model and dimensions of Box Dust Collector (BC) containment system.

**Figure 9 toxics-13-00957-f009:**
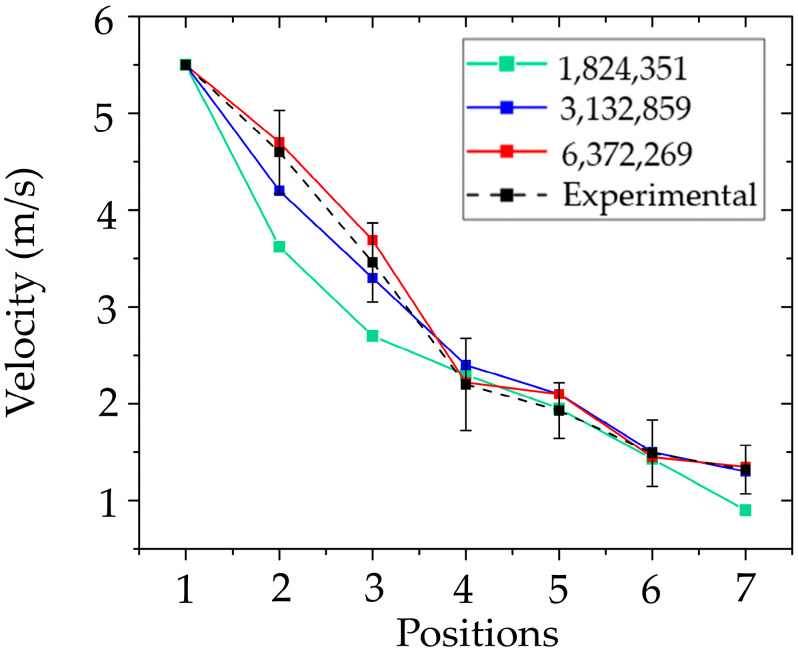
Mesh independence study and CFD model validation.

**Figure 10 toxics-13-00957-f010:**
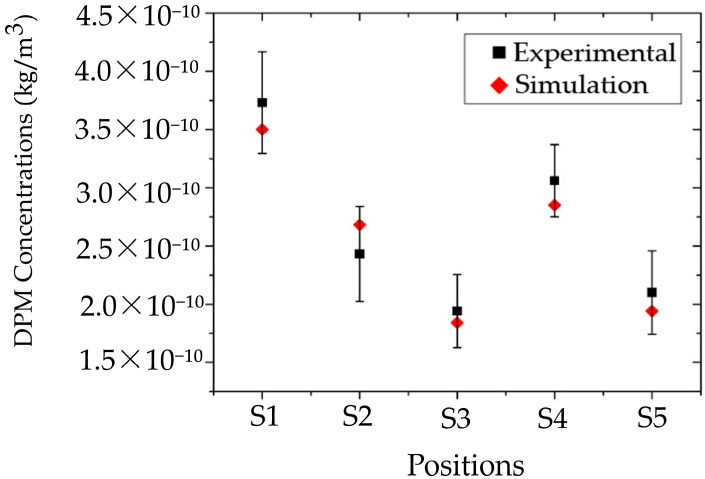
Validation of CFD model accuracy for Dispersed particulate matter (DPM) concentration.

**Figure 11 toxics-13-00957-f011:**
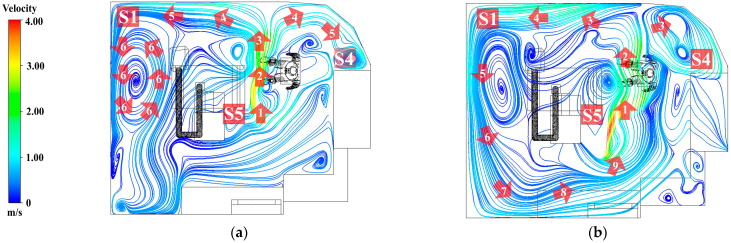
Airflow Streamlines and recirculation patterns in the operatory. (**a**) Airflow streamlines at the dentist’s working plane (0.45 m); (**b**) airflow streamline at the operator’s breathing zone (1.1 m).

**Figure 12 toxics-13-00957-f012:**
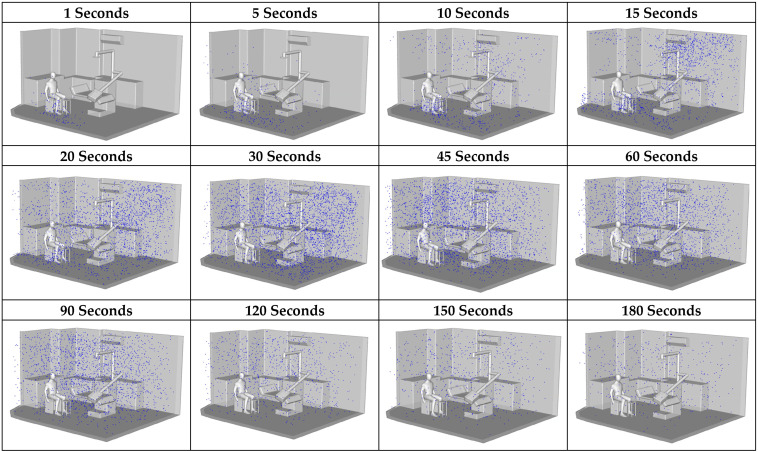
Particle dispersion effects according to different time intervals.

**Figure 13 toxics-13-00957-f013:**
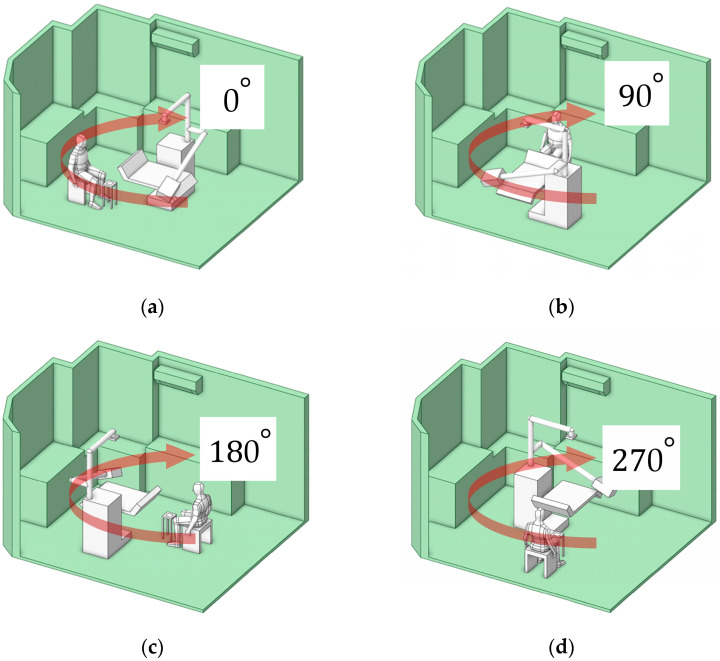
Four rotational orientations of working locations of the study. (**a**) 0°; (**b**) 90°; (**c**) 180°; (**d**) 270°.

**Figure 14 toxics-13-00957-f014:**
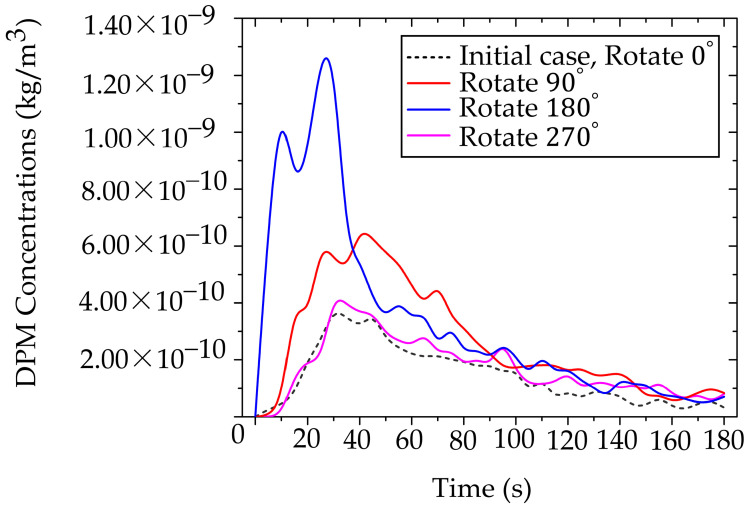
Quantitative analysis of occupational exposure across four working orientations.

**Figure 15 toxics-13-00957-f015:**
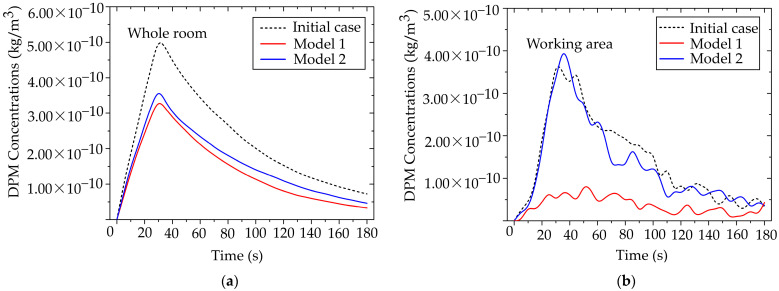
Comparative particulate mitigation efficacy of portable air cleaner model over time. (**a**) Concentration dynamics at the whole room scale, showing overall air cleansing efficiency; (**b**) concentration dynamics within the defined in working area.

**Figure 17 toxics-13-00957-f017:**
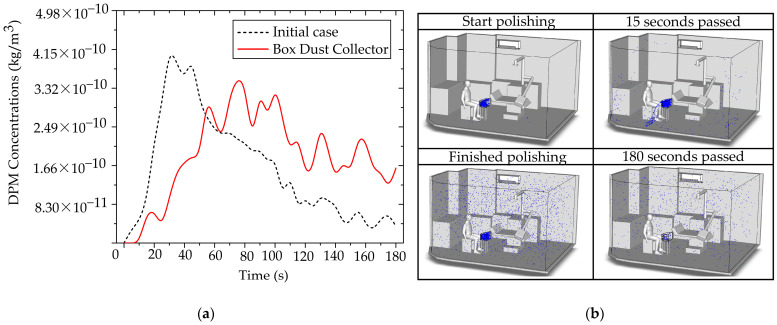
Box Dust Collector performance and particle leakage dynamics. (**a**) Comparison of DPM concentration within the dentist’s working zone for the initial case (0°) the BC case; (**b**) time-sequence visual series illustrating particle distribution.

**Table 2 toxics-13-00957-t002:** Specifications of measuring devices used for field measurement and CFD validation.

Instrument Name	Manufacturer	Model	Range	Flow Rate	Accuracy
Handheld Particle Counter	KANOMAX	3887	0.3–5 μm	0.1 cf/min 2.83 L/min	±10%
Air velocity meter	Testo	425	0 to +20 m/s	-	±5%

**Table 3 toxics-13-00957-t003:** Value of air velocity in measurement.

Positions	P1	P2	P3	P4	P5	P6	P7
Velocity of Air (m/s)	5.5 m/s	4.55 m/s	3.42 m/s	2.18 m/s	1.9 m/s	1.48 m/s	1.31 m/s

## Data Availability

Data are contained within the article.

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
