# Peer review of "Airborne Dental Material Particulates and Occupational Exposure: Computational and Field Insights into Airflow Dynamics and Control Strategies"

_toxics, 2025, doi:10.3390/toxics13110957_

Round 1
Reviewer 1 Report
Comments and Suggestions for Authors
This paper investigated occupational safety in a dental clinical room under various conditions using computational method. The method was well-established and validated against experimental measurements. This manuscript was well-written and presented a comprehensive study of particle behavior during operation using advanced CFD method. It provided valuable insights for researchers and engineers working on occupational micron particles exposure reduction. Considering it potentially represents a significant contribution to the field of occupational exposure study, I would recommend the publication of this manuscript after minor revision.
I only have a few minor comments for this manuscript:
- Line 141, no need to redefine abbreviations (computational fluid dynamics) - should do on first use and be consistent throughout.
- What were the time steps used for solving fluid flow and particles? What was the courant (CFL) number? Also, just out of curiosity, how much physical time did it take to complete one simulation case?
- Can the authors also provide the number of cells of the mesh in addition to the number of nodes?
Author Response
"Please see the attachment."

Reviewer 2 Report
Comments and Suggestions for Authors
This manuscript presents a timely and relevant investigation into airborne particulate exposure during dental procedures, combining CFD modeling with real-world measurements. The integration of engineering controls such as PACs and the BC is particularly valuable and provides practical recommendations for occupational safety. However, the paper would benefit from clearer justification of some methodological choices, a stronger discussion of health implications, and a more balanced integration of engineering findings with toxicological perspectives. Addressing these issues would significantly enhance the impact and clarity of the study.
-The introduction could more clearly distinguish between biological aerosols and non-biological particulates to highlight the novelty of the current study.
-The selection of 0.5 μm particles as the representative case should be justified with stronger references or comparative data.
-The assumed release rate and velocity of particles may not fully represent clinical variability; a sensitivity analysis would strengthen confidence in the findings.
-The validation process is thorough, but more discussion is needed on the limitations of the measuring instruments and their influence on accuracy.
-The discussion focuses heavily on CFD outcomes; more integration of toxicological or health risk perspectives would make the findings more relevant to occupational safety.
-The comparison of PAC models is valuable, but the generalizability to other clinic types, ventilation systems, or room sizes should be clarified.
-While the conclusions emphasize engineering strategies, potential barriers to implementation in actual dental practices (cost, space, ergonomics) should also be acknowledged.
-When discussing the adoption of PACs, please elaborate on the need to secure an appropriate Clean Air Delivery Rate (CADR) for the working environment. You may refer to the following paper for further guidance(https://doi.org/10.1155/2024/5055615).
-The future research section could more explicitly outline the need for studies on multiple dental materials and long-term cumulative exposure risks.
Author Response
"Please see the attachment."
